# Microbiological, Clinical and Radiological Aspects of Diabetic Foot Ulcers Infected with Methicillin-Resistant and -Sensitive *Staphylococcus aureus*

**DOI:** 10.3390/pathogens11060701

**Published:** 2022-06-17

**Authors:** Maria Stańkowska, Katarzyna Garbacz, Anna Korzon-Burakowska, Marek Bronk, Monika Skotarczak, Anna Szymańska-Dubowik

**Affiliations:** 1Department of Oral Microbiology, Medical Faculty, Medical University of Gdansk, 80-204 Gdansk, Poland; maria.stankowska@gumed.edu.pl; 22nd Division of Radiology, Medical Faculty, Medical University of Gdansk, 80-214 Gdansk, Poland; monika.skotarczak@gumed.edu.pl (M.S.); anna.szymanska-dubowik@gumed.edu.pl (A.S.-D.); 3Division of Preventive Medicine & Education, Medical Faculty, Medical University of Gdansk, 80-211 Gdansk, Poland; anna.korzon-burakowska@gumed.edu.pl; 4Laboratory of Clinical Microbiology, University Clinical Center, 80-952 Gdansk, Poland; msb@gumed.edu.pl

**Keywords:** diabetes, *Staphylococcus aureus*, methicillin resistance, MRSA, diabetic foot, DFU, X-ray

## Abstract

Diabetic foot ulcer (DFU) is one of the most common chronic complications of diabetes. This study aimed to assess the factors with an impact on the infection of diabetic foot ulcers by methicillin-resistant *S. aureus* and to evaluate the influence of methicillin resistance on the frequency of osteitis (based on classic X-ray images). A total of 863 patients suffering from DFU were analyzed during the study period. Out of 201 isolated *S. aureus* cases, 31 (15.4%) were methicillin-resistant (MRSA). MRSA infections were associated with a higher incidence of osteitis compared to MSSA infections (*p* << 0.0001), both the occurrence of smaller (<50%)) and greater (>50%) inflammatory bone changes (*p* << 0.0001). Furthermore, MRSA occurred significantly more frequently in men than in women (*p* < 0.01) and more often among patients with type 2 diabetes than among patients with type 1 diabetes (*p* < 0.05). MRSA were isolated statistically less often in overweight patients than in patients with normal BMI (*p* < 0.05). DFUs infected with MRSA were significantly more frequently associated with the presence of *Pseudomonas* sp. and other non-fermenting bacilli than those infected with MSSA (*p* < 0.05). To conclude, osteitis incidence is related to MRSA infection in patients with diabetic foot ulcers; thus, patients infected by *S. aureus* should be closely monitored in the course of using antibiotics and treated with narrow-spectrum antibiotics.

## 1. Introduction

Diabetic foot ulcer (DFU) is one of the chronic complications of diabetes and the International Working Group on the Diabetic Foot estimates that it annually affects approximately 26 million people with a global prevalence of 6.3% [1]. Diabetic foot ulcers are often complicated by bacterial infection. Superficial infection can spread quickly, involving subcutaneous tissue, muscles, joints and bones, leading to the onset of osteitis, which is one of the most feared complications of diabetes mellitus as it may result in amputation [2].

*Staphylococcus aureus*, considered one of the most significant etiological factors of DFU infection, has developed multiple mechanisms of antibiotic resistance which are transferred rapidly between the strains in both hospital and community settings [3]. The problem is particularly evident in the case of methicillin-resistant *S. aureus* (MRSA) which previously spread primarily in a hospital setting as hospital-acquired MRSA (HA-MRSA), but nowadays are increasingly found in a community setting as community-acquired MRSA (CA-MRSA), displaying high infectivity and virulence [4].

The medical literature usually makes no clear distinction between the terms osteitis and osteomyelitis. The main difference between osteitis and osteomyelitis is the way that the infection affects the bone, which is centripetal in osteitis and centrifugal in osteomyelitis. Because these terms are often used interchangeably in clinical settings, bone involvement by an inflammatory process in the diabetic foot will be referred to as osteitis here [5].

The aim of the study was to assess the factors with an impact on the infection of diabetic foot ulcers by methicillin-resistant *S. aureus* and to evaluate the influence of methicillin resistance on the frequency of osteitis (based on classic X-ray images). 

## 2. Results

### 2.1. Demographic and Clinical Study

A total of 201 DFU patients were enrolled during the study period. Of these, 150 (74.6%) were men. The largest age group was patients over 60 years of age, 132 (65.7%), and 69 (34.3%) patients were below 60 years of age. Type 2 diabetes was predominant among the patients (78.6%), and type 1 was less frequent (20.4%). There were two cases of type 3 diabetes. Type 3 diabetes mellitus has molecular and biochemical features that overlap with both type 1 and 2 diabetes mellitus, and therefore corresponds to a chronic insulin resistance plus insulin deficiency state. The largest group was made up of patients with long-term diabetes (>9 years) (79.6%). There were 42 (20.9%) patients with normal BMI, 54 (26.9%) were overweight and 105 were obese (56 had obesity class 1, 33 obesity class 2, and 16 obesity class 3). All patients were assessed for the level of glycated hemoglobin and CRP. The majority of diabetic patients had poor glycemic control with hemoglobin A_1c_ ≥ 7% (145—72.1%) and 56 (27.9%) were considered well-controlled with hemoglobin A_1c_ < 7%. Overall, 59.2% of people had elevated markers of inflammation in their blood at the diagnosis of diabetic foot ulcer infection (Table 1).

### 2.2. Isolated Bacteria

*S. aureus* was observed as a single species in 66 (32.8%) DFU samples. Gram-positive bacteria (71.1%) prevailed compared to Gram-negative bacteria (51.7%). Apart from *S. aureus*, Gram-positive cocci such as beta-hemolytic streptococci (groups A, B, C, G, and E), enterococci, and coagulase-negative staphylococci have been isolated. The second most common group of bacteria was made up of fermenting Gram-negative bacilli *Enterobacteriaceae*. Apart from *Enterobacteriaceae*, Gram-negative non-fermenting rods were detected, and *Pseudomonas aeruginosa* was the predominant bacterium, followed by *Acinetobacter baumanii*, *Pseudomonas putida*, *Stenotrophomonas maltophilia,* and *Alcaligenes faecalis*. Anaerobic non-spore- and spore-forming bacteria also were co-isolated, such as *Peptostreptococcus anaerobius*, *Bacteroides fragilis*, *Prevotella bivia*, *Prevotella melaninogenica*, *Veilonella parvula*, *Propionibacterium propionicum,* and *Clostridium sp.* (Table 2). In 135 (67.2%) cultures, polymicrobial growth (two or more bacterial species) was observed. Of these, 28 (13.7%) had more than two pathogens (Gram-negative rods predominated) (Table 3).

### 2.3. Antimicrobial Susceptibility of S. aureus Strains

Thirty-one (15.4%) of the analyzed *S. aureus* strains were methicillin-resistant (MRSA). Apart from resistance to β-lactams, the MRSA showed resistance to ciprofloxacin (93.5%), erythromycin (71%), clindamycin (71%), gentamicin (6.5%), and tetracycline (6.5%). Vancomycin-resistant strains were not detected. Seventy-one percent of MRSA were multidrug-resistant (MDR).

In the methicillin-susceptible *S. aureus* (MSSA) group, resistance to penicillin was demonstrated in 65.9% of cases, followed by resistance to erythromycin (29.4%), clindamycin (24.1%) and ciprofloxacin (8.2%). In this group, resistance to vancomycin, gentamicin and tetracycline was not detected. Multidrug resistance was observed among 17.6% of MSSA strains. 

### 2.4. MRSA versus MSSA DFU Infections

Diabetic foot wounds infected with MRSA were significantly more frequently associated with the presence of non-fermenting bacilli such as *Pseudomonas* sp. or *Acinetobacter* sp. than those infected with MSSA (OR = 3.4 [1.3–8.7], *p* < 0.05). For the remaining co-isolated bacteria, no significant differences were found (Table 4). 

The demographic and clinical features of patients with MSSA versus MRSA DFU infection are compared in Table 4. MRSA occurred significantly more frequently in men than in women (OR = 5.9 [1.3–25.6]; *p* < 0.01) and more often among patients with type 2 diabetes than among patients with type 1 diabetes (OR = 9.4 [1.2–70.9], *q* < 0.05). MRSA were isolated statistically less often in overweight patients than in patients with normal BMI (overweight: OR = 0.2 [0.1–0.6], *q* < 0.01; obesity class 1: OR = 0.1 [0.1–0.4], *q* < 0.0005; obesity class 2: OR = 0.1 [0.0–0.4], *q* < 0.0005; obesity class 3: OR = 0.2 [0.1–0.7], *q* < 0.05). There were no statistically significant differences in characteristics between patients with MSSA and MRSA infection in relation to age and diabetes duration. In both groups, diabetes duration was usually over 9 years. As many as two-thirds of patients with both MRSA and MSSA were above 60 years of age. In the MSSA group, elevated values of HbA1c and CRP dominated, while in the MRSA group the distribution of normal and elevated values of HbA1c and CRP was similar. Neither the HbA1c nor the CRP value was significantly associated with MRSA infection (Table 5).

### 2.5. Diabetic Foot Ulcers Complicated or Not with Osteitis

Out of the analyzed X-rays from 201 patients, 107 showed features characteristic of osteitis. Of these, 50 had less extent of inflammatory bone changes (<50%) and 57 were rated as having greater extent of changes (>50%). X-rays from 94 patients showed no changes characteristic of osteitis. MRSA infections were associated with a higher incidence of osteitis compared to MSSA infections (OR = 10.5 [3.1–35.9], *p* << 0.0001). A statistically significant difference was found for both the occurrence of smaller (<50%) (OR = 6.6 [1.7–25.6], *p* < 0.05) and greater (>50%) (OR = 14.6 [4.1–52.3], *p* << 0.0001) inflammatory bone changes (Figure 1, Table 6).

The analysis of bacteria co-isolated with *S. aureus* showed that the occurrence of osteitis may be related to the presence of *Actinomyces* sp. (*p* = 0.0620); however, this result did not reach the required level of statistical significance (*p* < 0.05). Therefore, it should be interpreted with caution. No statistically significant relationships were found between the occurrence of osteitis and other bacteria (Table 7). The presence of osteitis was significantly associated with the occurrence of resistance to fluoroquinolones (*p* < 0.0005) which is probably due to MRSA resistance to fluoroquinolones (Table 8). No statistically significant relationships were observed between the occurrence of osteitis and the demographic and clinical features of patients (Table 9).

## 3. Discussion

Diabetic foot disease is one of the most feared complications of diabetes mellitus. Foot ulcer may become complicated by infection and osteitis which may result in foot amputation. Successful treatment requires a holistic and multidisciplinary approach which should involve microbiological advice [6]. In this retrospective cohort study, we decided to assess the factors with an impact on the infection of diabetic foot ulcers by methicillin-resistant *S. aureus*.

*S. aureus* is one of the most common etiological factors of DFU infection. In the present study, apart from *S. aureus*, we observed a slight predominance of other Gram-positive bacteria, including β-hemolytic streptococci, enterococci, and coagulase-negative staphylococci. However, according to many authors, DFUs are often polymicrobial comprised of both Gram-positive and Gram-negative aerobic bacteria and anaerobes [7]. Our study similarly showed that the rate of polymicrobial infections was 67.2% compared to 32.8% of monomicrobial staphylococcal infections. The number of the etiological factors in DFU infection is associated with the duration of the ulcer and previous antimicrobial treatment. It was reported that in the early stages of infection the monomicrobial state prevails and as the infection progresses with time, a polymicrobial state arises [8].

The most important antibiotic resistance in staphylococci is methicillin resistance, which in clinical terms signifies resistance to all β-lactam antibiotics and is often accompanied by resistance to many other groups of antimicrobial agents [9]. In our study, the overall prevalence of methicillin-resistant *S. aureus* (MRSA) diabetic foot infection was 15.4%. The incidence of MRSA in our study population with DFU is similar to that in the literature and seems to be in the middle range of MRSA incidence observed in other countries. The reported prevalence of MRSA infections in DFU usually ranges between 10 and 30% with an alarming trend for an increase in many countries [10]. A recent study summarizing the incidence of MRSA in a group of 10,994 diabetic patients showed a similar MRSA prevalence rate at the level of 16.8% [11].

Knowledge about the local antibiotic susceptibility pattern of the pathogens is highly essential for the proper management of DFUs [12,13]. In the present study, isolated MRSA, as many as 93.5%, were resistant to ciprofloxacin, and above 70% were resistant to both erythromycin and clindamycin. Dissimilarly, vancomycin showed 100% effectiveness toward both MSSA and MRSA strains. This is consistent with the study by Rani et al., where the Gram-positive DFU pathogens showed complete sensitivity to vancomycin, linezolid, and teicoplanin [14]. Although all MRSA isolates in our study were fully susceptible to vancomycin, they were usually multiresistant, typical for MRSA strains. Aside from beta-lactams, multidrug-resistant strains most often showed resistance to macrolides, lincosamides, and fluoroquinolones, which carries the risk of failure in antibiotic treatment.

We analyzed the study group, divided into MSSA and MRSA infections, based on demographic and clinical features. In both groups, males were much more commonly involved. Such a distribution has been clearly shown in various studies in the literature as well [15,16]. Male sex is mentioned as one of the risk factors for the development of DFU among other factors such as older age [17]. In the present study, the majority (65.7%) of DFUs were seen predominantly in elderly people aged above 60 years. As many as two-thirds of MRSA and MSSA patients were above 60 years of age; these results are consistent with the current literature [18,19,20]. The increased prevalence among the elderly is due to multiple reasons such as a longer duration of diabetes, the presence of multiple comorbidities, and reduced immune status [21]. Most of the patients in our study have been suffering from diabetes for more than nine years. As is well known, diabetic foot syndrome is more common in patients with longstanding diabetes. As the duration of the disease increases, the chances of developing DFU also increase [17].

More than 78% of patients in our study have type 2 diabetes, which is not surprising. This type is the most common type of diabetes worldwide, being largely the result of excess body weight and physical inactivity [17,22]. Methicillin-resistant *S. aureus* (MRSA) is more often isolated from patients who have recently received antibiotic therapy, have been previously hospitalized, have a nasal carriage of MRSA or osteomyelitis, or have a long wound duration (≥4 weeks), which is more common in type 2 diabetes [17].

Most of the patients in the study group were overweight or obese (79.1%), of which patients infected with MRSA accounted for only 4.8%. Olsen et al. found a significant positive correlation between BMI and MRSA carriage in women, particularly among those aged 30–43 years [23]. The analysis of skin and soft tissue staphylococcal infections showed that obesity is related to the presence of methicillin resistance in staphylococci [24]. Despite these results, no such association was found in this study. Our study is not the first report of a lack of relation between obesity and increased frequency of infections [25,26]. Neidhart et al. reported a reduced risk of *S. aureus* carriage in obese (BMI ≥ 30.0 kg/m^2^) compared to overweight patients (BMI of 25.0 to 30 kg/m^2^) [27]. Our results shed new light on the clinical picture of a patient who may be at risk of a severe diabetic foot infection, which until now has been most often associated with obesity. With regard to the risk of osteitis, more attention should also be paid to patients whose weight is within the normal range. It seems that the relationship between obesity and DFI caused by MRSA is not clear and further studies are needed.

Over the past decade, hemoglobin A1C (HbA1C) was recognized as an indispensable parameter for the mid-term monitoring of glycemic control over a period of 2–3 months [28]. Poor glycemic control leads to the glycosylation of immune proteins which leaves patients more prone to infection [29,30]. Giurato et al. found a significant correlation between HbA1C levels and wound sizes, where patients with higher HbA1C levels usually had a larger wound size [31]. In our study, at the time of diagnosis of osteitis, 69.2% of patients had poorly controlled diabetes, which may confirm a global problem with diabetes control that is worth focusing on. According to recent studies, a 1% decrease in HbA1c results in a 21% reduction in all diabetic complications [32].

Overall, 63.6% of the patients had increased inflammatory blood markers at the diagnosis of bone infection. Osteitis manifestations are often very unspecific. Bone infection might be suspected by the presence of draining fistulas or in long-term wounds. Increased inflammatory markers may be of some help, although they are unspecific and show normal values in many cases as in our study [33]. In our study, 59.2% of patients had elevated markers of inflammation in their blood at diagnosis of diabetic foot ulcer with no statistically significant differences between the groups with and without osteitis.

Many studies reported the impact of methicillin-resistant *S. aureus* on increased time to wound healing, the need for surgical procedures, and the likelihood of treatment failure in patients with a diabetic foot infection. We found that MRSA infections were associated with a significantly higher incidence of osteitis compared to MSSA infections (*p* << 0.0001). This association provides evidence that methicillin resistance may affect the course of the infection. The patients who are diagnosed with MRSA infection should be more closely monitored to avoid consequences, even as extreme and not uncommon as amputation, from the very beginning of the diagnosis. Traditionally, MRSA bone and joint infections are considered more severe, leading to greater morbidity than those with methicillin-sensitive *S. aureus* [34,35,36]. An increased toxicity of MRSA as compared to MSSA strains has been also suggested [37,38]. However, there are conflicting studies about the role of methicillin resistance in the severity of infection. On the one hand, clinical data concerning the length of hospitalization, mortality rate and hospital costs related to MSSA and MRSA suggest a greater burden for MRSA infections [39]. In hospital settings, MRSA is also more commonly associated with bacteremia than MSSA, leading to a higher mortality rate. On the other hand, according to the World Health Organization (WHO), MRSA is generally not more virulent than MSSA [39]. The important reason for the conflicting results may be the non-homogeneous nature of *S. aureus* population. MRSA strains frequently consist of a heterogeneous population of bacterial cells, composed of methicillin-sensitive, borderline-resistant and methicillin-resistant (MR) subpopulations [40].

In the study, the presence of osteitis was not associated with the coexistence of bacteria other than *S. aureus*, except for the *Actinomyces* genera. However, this result did not reach the required level of statistical significance (*p* < 0.05). According to Gannepalli et al., the presence of *Actinomyces* sp. in DFU infections may increase the risk of osteitis [41]. Although *Actinomyces* sp. has a low virulence and invasion potential, its co-infection and the co-production of toxins or enzymes can lead to the development of osteitis. *Actinomyces* can act synergistically in forming an ecosystem with low oxidoreduction potential favorable for anaerobic bacteria growth. Bacteria destroy the highly vascularized aerobic system and replace it with a poorly irrigated granulated tissue thereby permitting an anaerobic environment [42,43]. *Actinomyces* sp. has been found to cause a sclerosing type of osteomyelitis mimicking bone tumors [42].

## 4. Materials and Methods

### 4.1. Study Population

A retrospective epidemiological study was performed on patients with diagnosed diabetic foot ulcers admitted to the Regional Diabetic Center of University Clinical Center in Gdansk between January 2017 to December 2019. The study was approved by the Independent Bioethics Committee for Scientific Research at the Medical University of Gdansk (NKBBN/520-232/2019). Infection was diagnosed according to the criteria proposed by the international consensus on the diabetic foot [44]. Only those DFU patients with microbial cultures positive for *S. aureus* were involved in this study. Another inclusion criterion was the presence of foot X-ray taken within 10–30 days from the diagnosis of ulcer infection.

A total of 863 patients with DFU were analyzed, and 389 patients were excluded because *S. aureus* was not cultured; for 173 patients, foot X-rays were not performed within 10–30 days from the diagnosis of ulcer infection. They were also excluded. In another 47 patients, the radiographic image was questionable and did not clearly indicate the presence or absence of osteitis. For 53 patients, some of the study’s relevant clinical data were missing in the documentation. These were not included. This resulted in 201 patients being analyzed for this study. We used the computer system at the Laboratory of Clinical Microbiology and Regional Diabetic Center of the University Clinical Center to retrieve information about the cases of *S. aureus* infection over this 3-year period.

The following parameters were evaluated: age at the time of admission to hospital, sex, medical history (diabetes mellitus type, BMI value), HbA1c value, and CRP value. Patients were labeled as overweight if their BMI was over 25, obese class 1 if BMI was over 30, obese class 2 > 35, and obese class 3 if more than 40 [45].

### 4.2. Bacterial Strains

This study was based on a retrospective microbiological analysis of cultures of specimens derived from DFU patients archived at the Laboratory of Clinical Microbiology, University Clinical Center in Gdansk, during routine clinical laboratory procedures. The scheme of microbiological diagnostics included the standard procedure of culture on microbiological media, macroscopic and microscopic evaluation of the cultured colonies, as well as serological and biochemical identification [46]. Briefly, samples were inoculated on enriched and selective media; Columbia agar with 5% defibrinated sheep blood, MacConkey, Chapman, bile and esculin medium (24 h incubation, 37°C, aerobic atmosphere), and Schaedler 5% Sheep Blood Agar and Vit K1 (incubated in an anaerobic atmosphere for 7 days at 37 °C). The cultured strains were identified by latex agglutination and biochemical tests (API ID strips) and, in case of doubtful identification, using matrix-assisted laser desorption/ionization-time of flight mass spectrometry (MALDI-TOF MS).

### 4.3. Antimicrobial Susceptibility

The antimicrobial susceptibility was determined on Mueller-Hinton agar plates (Becton Dickinson, Franklin Lakes, NJ, USA) by the disk diffusion method and interpreted according to the EUCAST [47]. The following antimicrobial agents were tested: oxacillin, cefoxitin, gentamicin, erythromycin, clindamycin, tetracycline, ciprofloxacin, amoxicillin/clavulanic acid (Bio-Rad, Marnes la Coquette, France) and penicillin G (Oxoid, Basingstoke, UK). The inducible resistance to macrolide-lincosamide-streptogramin B (MLS_B_) was detected by disk diffusion method with clindamycin (2 μg) and erythromycin (15 μg) disks positioned 15–26 mm apart [47]. MIC for vancomycin was determined by E-tests, in line with the manufacturer’s instructions (AB Biodisc, Solna, Sweden).

Resistance to methicillin was first identified using cefoxitin (30 µg) and oxacillin (1 µg) disks, and then confirmed by the detection of PBP2a protein (OXOID ™ PBP2 ’Latex Agglutination Test Kit, Basingstoke, UK).

Multidrug resistance (MDR) was defined as a resistance to three or more classes of antimicrobials.

### 4.4. Osteitis and X-ray Evaluation

The presence of osteitis and its extent was identified after a systematic review of the patient records (focusing on the identification of active bone changes in the course of inflammation). Individual patient’s x-ray was checked by a minimum of two observers for radiological markers of active inflammation such as the presence of soft tissue swelling, periosteal new bone formation, loss of trabecular architecture, cortical bone destruction, focal osteopenia or permeative radiolucency [5,48,49,50]. According to the radiological assessment, the study group was divided into patients with and without osteitis. In the group of patients with osteitis, the percentage of bone width affected by the inflammation was determined based on the image in the AP (anterior–posterior) projection. The extent of bone inflammation was divided into two groups: <50% and >50% of the affected bone area.

### 4.5. Statistical Analysis

All categorical data are presented as counts and/or frequencies (percentages). Statistical significance of sample between-group differences was assessed by means of the exact Fisher’s or Fisher–Freeman–Halton tests, depending on the dimension of respective contingency table (2 × 2 or larger, respectively). Multiple-comparison *post hoc* testing was performed using the exact Fisher’s test with the false discovery rate being controlled for by the *FDR* correction according to Benjamini and Hochberg [51]. Effect sizes were expressed by means of odds ratio coefficients (OR) with respective 95% confidence intervals (95% CI). Statistical significance was inferred for *p* < 0.05.

All statistical analyses were performed in R ver. 4.0.4 (Vienna, Austria) [52].

## 5. Conclusions

In conclusion, MRSA were prevalent in 15.4% of patients with diabetic foot ulcer and showed resistance to common antimicrobial agents. We found some significant differences in clinical and radiological features in patients with isolated MSSA and MRSA from DFUs. The most important difference was in the frequency of osteitis. MRSA infections were associated with a higher incidence of osteitis compared to MSSA infections. This effect seems to apply to both the occurrence of smaller (<50%) and greater (>50%) inflammatory bone changes. Furthermore, MRSA occurred significantly more frequently in men than in women and more often among patients with type 2 diabetes than among patients with type 1 diabetes. MRSA were isolated statistically less often in overweight patients than in patients with normal BMI. DFUs infected with MRSA were significantly more often associated with the presence of *Pseudomonas* sp. and other non-fermenting bacilli than those infected with MSSA. To conclude, osteitis is related to MRSA infection in diabetic foot ulcers; hence, patients with *S. aureus* infection should be closely monitored in the course of using antibiotics and treated with narrow-spectrum antibiotics.

## Figures and Tables

**Figure 1 pathogens-11-00701-f001:**
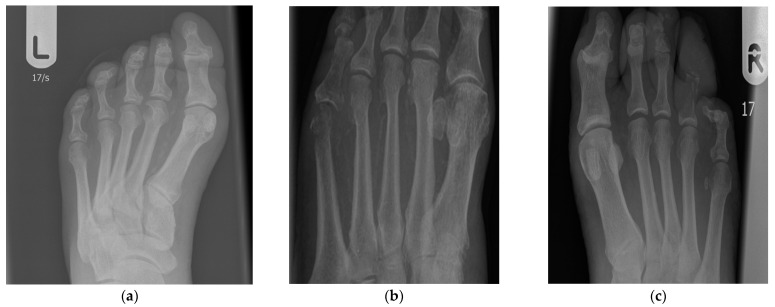
(**a**) X-ray showing hard and smooth bone of a DFU patient that appears normal and not infected. Findings of early inflammatory process may be subtle and include soft tissue changes, muscle swelling, and blurring of the soft tissue planes; here, a slight thickening of the soft tissues around the phalanges of the fourth toe can be seen. Calcifications of the interdigital arteries observed in this X-ray are rarely seen in people without diabetes; (**b**) X-ray showing circumscribed loss of well-defined cortical contours observed at the lateral aspect of the fifth metatarsal head corresponding to the location of the ulcer; inflammatory process involving less than 50% of the bone width; (**c**) X-ray showing almost complete destruction (involving more than 50% of the bone width) of the middle and distal phalanges of the fourth toe; slightly less intense changes are also visible in the proximal phalanx.

**Table 1 pathogens-11-00701-t001:** Characteristics of population study suffering from diabetic foot ulcer (DFU) infected with *S. aureus*.

Variable		Number of Patients with DFU Infected *S. aureus* (n = 201)	Percentage (%)
Sex	Female	51	25.4
	Male	150	74.6
Age (years)	<40 years	3	1.5
	40–60 years	66	32.8
	>60 years	132	65.7
Diabetes type	1	41	20.4
	2	158	78.6
	3	2	1.0
Diabetes duration	<5 years	12	5.9
	5–9 years	29	14.4
	>9 years	160	79.6
BMI (kg/m^2^)	Normal weight	42	20.9
	Overweight	54	26.9
	Obesity class 1	56	27.9
	Obesity class 2	33	16.4
	Obesity class 3	16	7.9
HbA_1c_ level (mmol/mol)	<7	56	27.9
	7.1–7.9	35	17.4
	≥8	110	54.7
CRP (mg/L)	<5	82	40.8
	5–99	94	46.8
	≥99	25	12.4

**Table 2 pathogens-11-00701-t002:** Distribution of bacteria co-isolated with *S. aureus* from diabetic foot ulcer.

Bacteria Co-Isolated with *S. aureus*	Number	Percentage (%)
Only *Staphylococcus aureus*	66	32.8
Gram-positive cocci		
Coagulase negative staphylococci		
*Staphylococcus epidermidis*	6	3
*Staphylococcus haemolyticus*	5	2.5
*Staphylococcus capitis*	3	1.5
*Staphylococcus hominis*	1	0.5
*Staphylococcus simulans*	1	0.5
Beta-hemolytic streptococci		
group A	2	1
group B	16	8
group C	6	3
group G	13	6.5
group E	1	0.5
Alpha-hemolytic streptococci	4	2
*Enterococcus faecalis*	12	6
*Enterococcus faecium*	6	3
*Enterococcus durans*	1	0.5
Gram-negative bacilli		
*Enterobacteriaceae*		
*Escherichia coli*	22	11
*Proteus mirabilis*	16	8
*Proteus vulgaris*	2	1
*Enterobacter cloacae*	12	6
*Enterobacter hermannii*	1	0.5
*Morganella morganii*	5	2.5
*Klebsiella pneumoniae*	5	2.5
*Klebsiella oxytoca*	4	2
*Citrobacter koseri*	3	1.5
*Citrobacter freundii*	3	1.5
*Serratia marcescens*	5	2.5
*Raoutella planticola*	1	0.5
Non-fermenting bacilli		
*Pseudomonas aeruginosa*	17	8.5
*Pseudomonas putida*	1	0.5
*Acinetobacter baumannii*	5	2.5
*Stenotrophomonas maltophilia*	1	0.5
*Alcaligenes faecalis*	1	0.5
Anaerobes non spore-forming		
*Peptostreptococcus anaerobius*	7	3.5
*Bacteroides fragilis*	4	2
*Prevotella bivia*	2	1
*Prevotella melaninogenica*	1	0.5
*Veillonella parvula*	1	0.5
*Propionibacterium propionicum*	1	0.5
Anaerobes spore-forming		
*Clostridium sp.*	32	15.9
*Actinomyces sp.*	5	2.5

**Table 3 pathogens-11-00701-t003:** Distribution of bacteria co-isolated with S. aureus from diabetic foot ulcers in polymicrobial infections.

One Bacterium Co-Isolated with *S. aureus*	Two Bacteria Co-Isolated with *S. aureus*	Three Bacteria Co-Isolated with *S. aureus*	n (%) (n = 135)
*Enterobacteriaceae*			37 (27.4)
Beta-hemolytic streptococci			13 (9.6)
Enterococci			5 (3.7)
Spore-forming anaerobes			3 (2.2)
Coagulase negative staphylococci			2 (1.5)
Non-spore-forming anaerobes			2 (1.5)
Alpha-hemolytic streptococci			2 (1.5)
Non-fermenting bacilli			0 (0)
*Actinomyces sp.*			0 (0)
*Enterobacteriaceae*	Anaerobes spore-forming		11 (8.2)
Non-fermenting bacilli	Coagulase negative staphylococci		7 (5.2)
*Enterobacteriaceae*	Non-fermenting bacilli		7 (5.2)
*Enterobacteriaceae*	Coagulase negative staphylococci		6 (4.4)
*Enterobacteriaceae*	Beta-hemolytic streptococci		4 (3)
Beta-hemolytic streptococci	Non-fermenting bacilli		4 (3)
*Enterobacteriaceae*	Alpha- hemolytic streptococci		2 (1.5)
Beta-hemolytic streptococci	*Actinomyces* sp.		1 (0.7)
*Enterobacteriaceae*	*Actinomyces* sp.		1 (0.7)
Beta-hemolytic streptococci	Enterococci	Anaerobes non-spore-forming	8 (5.9)
Non-fermenting bacilli	Beta-hemolytic streptococci	Anaerobes spore-forming	7 (5.2)
*Enterobacteriaceae*	Anaerobes spore-forming	Anaerobes non-spore-forming	6 (4.4)
*Enterobacteriaceae*	Enterococci	Anaerobes spore-forming	4 (3)
*Enterobacteriaceae*	Enterococci	*Actinomyces* sp.	1 (0.7)
Beta-hemolytic streptococci	Enterococci	*Actinomyces* sp.	1 (0.7)
Spore-forming anaerobes	Coagulase negative staphylococci	*Actinomyces* sp.	1 (0.7)

**Table 4 pathogens-11-00701-t004:** Distribution of bacteria co-isolated with methicillin-sensitive (MSSA) and -resistant *S. aureus* (MRSA) infected with diabetic foot ulcer.

Bacteria Co-Isolated with *S. aureus*	MSSA	MRSA	Total	*p*-Value	OR [95% CI]
	(n = 170)	(n = 31)	(n = 201)		
Gram-positive cocci					
Coagulase negative staphylococci	13	3	16	0.7181	1.29 [0.35–4.84]
Streptococci	38	4	42	0.3367	0.51 [0.17–1.56]
Enterococci	13	6	19	0.0510	2.90 [1.01–8.33]
Gram-negative bacilli					
*Enterobacteriaceae*	65	15	80	0.3217	1.51 [0.70–3.27]
Non-fermenting bacilli	16	8	24	0.0162	3.35 [1.29–8.70]
Anaerobes					
Non-spore-forming anaerobes	13	6	19	0.0510	2.90 [1.01–8.33]
Spore-forming anaerobes	30	2	32	0.1796	0.32 [0.07–1.42]
*Actinomyces* sp.	5	0	5	0.5996	0.00 [0.00–NaN]

**Table 5 pathogens-11-00701-t005:** Characteristics of population study with diabetic foot ulcer infected with methicillin-sensitive and -resistant *S. aureus*.

Variable		Number of Patients with DFU Infected *S. aureus* (n = 201)	MSSA Infection (n = 170)	MRSA Infection (n = 31)	*p*-Value
Sex	Female	51	49	2	0.0067
	Male	150	121	29	
Age (years)	<40 years	3	2	1	0.4705
	40–60 years	66	57	9	
	>60 years	132	111	21	
Diabetes type	1	41	40	1	0.0157
	2	158	128	30	
	3	2	2	0	
Diabetes duration	<5 years	12	10	2	0.1196
	5–9 years	29	28	1	
	>9 years	160	132	28	
BMI (kg/m^2^)	Normal weight	42	24	18	<<0.0001
	Overweight	54	46	8	
	Obesity class 1	56	54	2	
	Obesity class 2	33	33	0	
	Obesity class 3	16	13	3	
HbA_1c_ level (mmol/mol)	<7	56	42	14	0.0835
	7.1–7.9	35	31	4	
	≥8	110	97	13	
CRP (mg/L)	<5	82	66	16	0.1647
	5–99	94	80	14	
	≥99	25	24	1	

**Table 6 pathogens-11-00701-t006:** Prevalence of osteitis in diabetic foot ulcer infected with methicillin-sensitive and -resistant *S. aureus*.

	MSSA (n = 170)	MRSA (n = 31)	*p*-Value	OR [95% CI]
Osteitis (−) (n = 94)	91	3	<<0.0001	10.5 [3.075–35.858]
Osteitis (+) (n = 107)	79	28		
osteitis <50% (n = 50)	41	9	0.012	6.585 [1.694–25.603]
osteitis >50% (n = 57)	38	19	<<0.0001	14.615 [4.087–52.269]

**Table 7 pathogens-11-00701-t007:** Distribution of bacteria co-isolated with *S. aureus* from diabetic foot ulcer complicated or not with osteitis.

Bacteria Co-Isolated with *S. aureus*	Osteitis (+)	Osteitis (−)	Total	*p*-Value	OR [95% CI]
	(n = 107)	(n = 94)	(n = 201)		
Gram-positive cocci					
Coagulase negative staphylococci	7	9	16	0.4472	0.66 [0.24–1.85]
streptococci	23	19	42	0.8633	1.08 [0.55–2.14]
Enterococci	14	5	19	0.0891	2.68 [0.93–7.75]
Gram-negative bacilli					
*Enterobacteriaceae*	45	35	80	0.5638	1.22 [0.69–2.16]
Non-fermenting bacilli	13	11	24	1.0000	1.04 [0.44–2.45]
Anaerobes					
Non spore-forming anaerobes	13	6	19	0.0510	2.9 [1.01–8.33]
Spore-forming anaerobes	16	16	32	0.7038	0.86 [0.40–1.83]
*Actinomyces sp.*	5	0	5	0.0620	Inf [NaN–Inf]

Inf—infinite values; NaN—Not A Number, undefined value.

**Table 8 pathogens-11-00701-t008:** Resistance to antibiotics of *S. aureus* infected diabetic foot ulcer complicated or not with osteitis.

Antibiotics Group Resistance	Osteitis (+)	Osteitis (−)	*p*-Value	OR [95% CI]
	(n = 107)	(n = 94)		
Beta-lactams	74	69	0.0001	0.32 [0.18–0.57]
Macrolides	32	40	0.0769	0.58 [0.32–1.03]
Lincosamides	30	33	0.2907	0.72 [0.40–1.31]
Fluoroquinolones	33	10	0.0005	3.75 [1.73–8.12]
Aminoglycosides	2	0	0.4996	Inf [NaN–Inf]
Tetracyclines	2	0	0.4996	Inf [NaN–Inf]
Glycopeptides	0	0	1.0000	Inf [NaN–Inf]

Inf—infinite values; NaN—Not A Number, undefined value.

**Table 9 pathogens-11-00701-t009:** Characteristics of patients with diabetic foot ulcer complicated or not with osteitis.

Variable		Number of Patients with DFU Infected *S. aureus* (n = 201)	Osteitis (+) (n = 107)	Osteitis (−) (n = 94)	*p*-Value
Sex	Female	51	23	28	0.1963
	Male	150	84	66	
Age (years)	<40 years	3	1	2	0.8603
	40–60 years	66	35	31	
	>60 years	132	71	61	
Diabetes type	1	41	20	21	0.4969
	2	158	85	73	
	3	2	2	0	
Diabetes duration	<5 years	12	6	6	0.9644
	5–9 years	29	15	14	
	>9 years	160	86	74	
BMI (kg/m^2^)	Normal weight	42	35	7	0.0002
	Overweight	54	28	26	
	Obesity class 1	56	23	33	
	Obesity class 2	33	13	20	
	Obesity class 3	16	8	8	
HbA_1c_ level (mmol/mol)	<7	56	33	23	0.4886
	7.1–7.9	35	16	19	
	≥8	110	58	52	
CRP (mg/L)	<5	82	39	43	0.1038
	5–99	94	50	44	
	≥99	25	18	7	

## Data Availability

Not applicable.

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
