# Peer review of "Microbiological, Clinical and Radiological Aspects of Diabetic Foot Ulcers Infected with Methicillin-Resistant and -Sensitive Staphylococcus aureus"

_pathogens, 2022, doi:10.3390/pathogens11060701_

Round 1

Reviewer 1 Report

Unfortunately, the authors did not address my previous major concerns. In my previous review, I have requested the authors to discuss the polymicrobial infection and rewrite the discussion portion. This has not been done.

"It is understandable that there were 67.2% of polymicrobial infection, how many among them had more than 2 pathogen? and what were those pathogens?" 

I was expecting a modification in table 2 to include the detailed result. Please try to make a table that enables readers to understand how many patients had S. aureus, how many had S. aureus + Pseudomonas and How many had S. aureus + Pseudonomas + Clostridium etc? 

In response to my comment "in the discussion section authors focus on what other researchers found but fail to write the implication of their findings." the authors added some texts towards the end of discussion, but this is still not enough. Please try to discuss and highlight the importance/findings of this study.

The mechanisms of antibiotic resistance is not methicillin resistance. Methicillin is antibiotic. Please rewrite the sentences.

Author Response

Dear Reviewer,

First, we would like to thank you for a detailed review of our manuscript and all of your valuable remarks. We have addressed them all in detail below. We hope that you will find our explanations and manuscript modifications sufficient for reconsidering its publication in the Pathogens.

With kind regards,

Authors

Response to Reviewer 1 Comments

I was expecting a modification in table 2 to include the detailed result. Please try to make a table that enables readers to understand how many patients had S. aureus, how many had S. aureus + Pseudomonas and How many had S. aureus + Pseudonomas + Clostridium etc? 

Re: New Table 3 has been added in line with Reviewer’s suggestion.

In response to my comment "in the discussion section authors focus on what other researchers found but fail to write the implication of their findings." the authors added some texts towards the end of discussion, but this is still not enough. Please try to discuss and highlight the importance/findings of this study.

Re: The importance of our study has been discussed in the revised text, lines 238-240, 278-282, and 303-307.

The mechanisms of antibiotic resistance is not methicillin resistance. Methicillin is antibiotic. Please rewrite the sentences.

Re: The sentence has been rewritten, line 221.

Reviewer 2 Report

Microbiological, clinical and radiological aspects of diabetic foot ulcers infected with methicillin-resistant and -sensitive Staphylococcus aureus

In the manuscript entitled “Microbiological, clinical and radiological aspects of diabetic foot ulcers infected with methicillin-resistant and -sensitive Staphylococcus aureus” the authors have carried out a retrospective cohort study to find any association between different clinical parameters and MRSA infection in diabetic foot ulcer patients. They have also studied the influence of methicillin resistance on the incidence of osteitis. The authors have carried out extensive morphological and culture dependent analysis to identify the co-occurring bacteria in infected DFU. However, the observations drawn on the radiological aspect is based only on x-ray images and lacks any other analysis. Overall, the article is well written and provides useful information for the clinicians that can be used for the monitoring and treatment of patients with S. aureus infected DFUs.

Some minor points:

L19: “201 isolates of”

L26: It is suggested that details of statistical significance can be described in the results rather than in the abstract.

L44: has developed

L54: referred to as

L60: a total of 201 DFU patients with S. aureus infection – right?

L62: Replace ‘had less than 60 years’ with ‘were below 60 years of age’

L63: define type 3 diabetes to make it clearer to the audience

L84: Acinetobacter baumanii, Pseudomonas putida, Stenotrophomonas maltophilia, and Alcaligenes faecalis

L95: In the methicillin-susceptible S. aureus (MSSA) group,

L97: In this group,

L105: patients with MSSA versus MRSA DFU

L115: In the MSSA group,

L124: Section has no text and is abrupt.

L141: as having greater

L142-145: Rephrase the sentence to make it clearer.

L143: How were the values 26% vs 3% arrived at?

L157: Since the data has overlapping data points, can this be represented in a better way?

L158: Define Inf and NaN in caption

L169: Beta hemolytic

L170: DFI, DFU and DFS are used interchangeably

L177: Methicillin resistance is not a mechanism.

L194: typical of

L211: received antibiotic therapy, have been previously hospitalized,

L238: Omit 'overall'

L314: Omit 'use the'

L320: If there is resistance to more than one antibiotic but both belonging to same class of antimicrobial?

L336: Change simple to sample

L354: In Line 147, authors postulate that occurrence of osteitis may be related to the presence of Actinomyces sp. based on its presence in osteitis positive vs negative cases. However, they did not find it to be statistically significant. Why only a select species was checked? Did the authors check for possible association with Pseudomonas? (Since there is a significant association of MRSA with Pseudomonas)

L357: foot ulcers; hence, patients

Table 1, 4 and 8 data can be made into one table.

Table 5- Sample size of MRSA (n=170) and MSSA (n=31) are skewed, which makes it difficult to compare and arrive at any valuable conclusions. Is it justified to conclude that MRSA infections were associated with significantly higher incidence of osteitis compared to MSSA infections, considering the huge difference in the sample size.

Table 8: Title has to be rephrased

Author Response

Dear Reviewer,

First, we would like to thank you for a detailed review of our manuscript and all of your valuable remarks. We have addressed them all in detail below. We hope that you will find our explanations and manuscript modifications sufficient for reconsidering its publication in the Pathogens.

With kind regards,

Authors

Response to Reviewer 2 Comments

Overall, the article is well written and provides useful information for the clinicians that can be used for the monitoring and treatment of patients with S. aureus infected DFUs.

Some minor points:

L19: “201 isolates of”

Re: The phrase has been changed, line 19.

L26: It is suggested that details of statistical significance can be described in the results rather than in the abstract.

Re: The details of statistical significance has been removed, in the Abstract.

L44: has developed

Re: It has been corrected, line 41.

L54: referred to as

Re: It has been corrected, line 61.

L60: a total of 201 DFU patients with S. aureus infection – right?

Re: Yes, it is correct.

L62: Replace ‘had less than 60 years’ with ‘were below 60 years of age’

Re: The phrase has been changed, line 70.

L63: define type 3 diabetes to make it clearer to the audience

Re: Type 3 diabetes has been defined, lines 72-74.

L84: Acinetobacter baumanii, Pseudomonas putida, Stenotrophomonas maltophilia, and Alcaligenes faecalis

Re: It has been corrected, line 93.

L95: In the methicillin-susceptible S. aureus (MSSA) group,

Re: It has been corrected, line 120.

L97: In this group,

Re: It has been corrected, line 122.

L105: patients with MSSA versus MRSA DFU

Re: It has been corrected, line 137.

L115: In the MSSA group,

Re: It has been corrected, line 147.

L124: Section has no text and is abrupt.

Re: The section contains only representative X-ray images and their description (Figure 1).

L141: as having greater

Re: It has been changed, line 172.

L142-145: Rephrase the sentence to make it clearer.

Re: The sentence has been rephrased, lines 174-175.

L143: How were the values 26% vs 3% arrived at?

Re: This confusion has been deleted.

L157: Since the data has overlapping data points, can this be represented in a better way?

Re: The authors tried to present the data most compactly and understandably possible.

L158: Define Inf and NaN in caption

Re: The abbreviations have been defined, lines 195, 198.

L169: Beta hemolytic

Re: It has been corrected, line 213.

L170: DFI, DFU and DFS are used interchangeably

Re: The DFI and DFS have been replaced with DFU throughout the revised manuscript.

L177: Methicillin resistance is not a mechanism.

Re: It has been corrected, line 221.

L194: typical of

Re: It has been changed, line 238.

L211: received antibiotic therapy, have been previously hospitalized,

Re: It has been corrected, line 268.

L238: Omit 'overall'

Re: The ward has been deleted.

L314: Omit 'use the'

Re: The wards have been deleted.

L320: If there is resistance to more than one antibiotic but both belonging to same class of antimicrobial?

Re: Yes, but that has not been qualified as multi-drug resistance.

L336: Change simple to sample

Re: It has been corrected, line 399.

L354: In Line 147, authors postulate that occurrence of osteitis may be related to the presence of Actinomyces sp. based on its presence in osteitis positive vs negative cases. However, they did not find it to be statistically significant. Why only a select species was checked? Did the authors check for possible association with Pseudomonas? (Since there is a significant association of MRSA with Pseudomonas)

Re: All bacteria have been checked according to Table 4.

L357: foot ulcers; hence, patients

Re: It has been changed, line 422.

Table 1, 4 and 8 data can be made into one table.

Re: The authors understand that some data is repeated in the Tables 1, 4 and 8 but this arrangement of Tables (three, not one) allows for presenting the results in the planned order.

Table 5- Sample size of MRSA (n=170) and MSSA (n=31) are skewed, which makes it difficult to compare and arrive at any valuable conclusions. Is it justified to conclude that MRSA infections were associated with significantly higher incidence of osteitis compared to MSSA infections, considering the huge difference in the sample size.

Re: The authors are aware that the small proportion of MRSA strains compared to the number of MSSA strains has been a limitation in this study but the statistical analysis used allows us to draw such conclusions and the authors hope that our findings offer a valuable insight into the S. aureus infection in patients with diabetic foot ulcers. 

Table 8: Title has to be rephrased

Re: The title has been rephrased, line 199.

Reviewer 3 Report

The manuscript "Microbiological, clinical and radiological aspects of diabetic foot ulcers infected with methicillin-resistant and -sensitive Staphylococcus aureus" is interesting, well-described, and presents relevant data. The study aimed to assess the factors with impact on the infection of diabetic foot ulcers by methicillin-resistant S. aureus and to evaluate the influence of methicillin resistance on the frequency of osteitis. The manuscript appears to have already undergone a revision, which immensely helped to remedy possible deficiencies.

However, a few details I think are necessary to be considered before approval for publication:

1) Remove the phrase from the results: S. aureus was present in all clinical specimens. Only positive patients for S. aureus entered the analysis, so this sentence becomes unnecessary.

2) The authors mention that 28 (13.7%) had more than two pathogens. However, in table 2, the microorganisms were considered individually. It would also be interesting to describe the polymicrobial that occurred. In addition, this polymicrobial presence should be considered for MRSA and osteitis correlation analyses. A more in-depth discussion of this subject deserves to be considered.

Author Response

Dear Reviewer,

First, we would like to thank you for a detailed review of our manuscript and all of your valuable remarks. We have addressed them all in detail below. We hope that you will find our explanations and manuscript modifications sufficient for reconsidering its publication in the Pathogens.

With kind regards,

Authors

Response to Reviewer 3 Comments

However, a few details I think are necessary to be considered before approval for publication:

1) Remove the phrase from the results: S. aureus was present in all clinical specimens. Only positive patients for S. aureus entered the analysis, so this sentence becomes unnecessary.

Re: The phrase has been removed.

2) The authors mention that 28 (13.7%) had more than two pathogens. However, in table 2, the microorganisms were considered individually. It would also be interesting to describe the polymicrobial that occurred. In addition, this polymicrobial presence should be considered for MRSA and osteitis correlation analyses. A more in-depth discussion of this subject deserves to be considered.

Re: New Table 3 described polymicrobial infection has been added in line with Reviewer’s suggestion. We appreciate the suggestion from the Reviewer to take into account other factors but the importance of the other etiological factors than S. aureus was out of the scope of this study. The main objective of this paper was to assess the infection of S. aureus.

Round 2

Reviewer 1 Report

This revision looks good to publish.

This manuscript is a resubmission of an earlier submission. The following is a list of the peer review reports and author responses from that submission.

Round 1

Reviewer 1 Report

In this manuscript titled "Microbiological, clinical and radiological aspects of diabetic foot ulcers infected with methicillin-resistant and -sensitive Staphylococcus aureus", the authors report a retrospective epidemiological study of 863 DFU patients with the aim to understand the factors that contribute to DFU infection by MRSA. Their results highlighted that osteitis incidence was higher in case of MRSA infections of DFUs.

The study is well thought out and executed. The methods are appropriate and results support the conclusions. The manuscript is well written. 

There are no concerns with this manuscript.

Reviewer 2 Report

The manuscript by Stańkowska et al., analysis diabetes foot ulcer associated with S. aureus infection. The manuscript has enough sample size and analysis to support the results. The manuscript can be presented in much better way to be more informative. The quality of presentation and the discussion of key important findings must be improved. For instance, in the discussion section authors focus on what other researchers found but fail to write the implication of their findings. Next, polymicrobial infection needs to be explained in detail. It is understandable that there were 67.2% of polymicrobial infection, how many among them had more than 2 pathogen? and what were those pathogens?

I have some minor comments too:

  1. while writing significance, I see the authors used <<, please clarify if it is a typo or has some meaning.
  2. Line 179 The most important mechanism of resistance in staphylococci is methicillin resistance. What is the meaning of the word resistance in this sentence?
  3. Sometimes, decimals are indicated by comma, please correct

Reviewer 3 Report

The article by StaÅ„kowska et al. analyzed the factors impacting the foot infections due to MRSA in persons living with diabetes. The retrospective study was conducted in a University Clinical Center during 3 years.  

Globally, this paper does not bring any scientific novelty. Moreover several important limitations could be noted limiting the impact of this paper.

-1. The methodology to assess the main objective of this paper is not adapted: the authors identified S. aureus and after they concluded to the infection… The presence of S. aureus is not equal to infection…. The diabetic foot infections (DFI) or diabetic foot osteomyelitis (DFOM) must be clinically diagnosed followed by the use of biological markers and radiology and after an adapted sampling. This is an important problem in this paper because the authors concluded to infection in 40.8% of patients harboring negative CRP. Publications consistently showed that CRP values are significantly higher in the presence of DFI than in diabetic foot ulcers (Uzun G et al., Tohoku J Exp Med 2007; Park JH et al., Diabetes Res Clin Pract 2017; Jeandrot A et al., Diabetologia 2008).To my point of view, the wounds of these patients are only colonised… this constitute a major bias in the data interpretation.

-2. The international classification (provided by the IWGDF consensus) used to grade the DFI must be obligatorily used in this paper.

-3. There is no presentation of the sampling methods. During the 3-year period, could the authors be sure that there are no modification in the protocol (give references?) ? Do the authors used systematically debridement before each sample? What is the protocol for bone biopsy ? A suspicion of osteitis needs the bone biopsy to corroborate the diagnosis. In this paper, I have a doubt on this approach… How the samples have been obtained ? Please explained the repartition of S. aureus following the sampling methods used. I assume that the authors know that swabs are particularly not adapted for deep samples with notably a limitation in the identification of all microorganisms present in the site of infection. The authors must indicate the proportion of each sampling methods and compared each other with the repartition of MSSA and MRSA. If MRSA was isolated from swabs in DFOM, this microorganism can be involved in the infection (please read Senneville E et al., Clin Infect Dis 2009).

-4. Bacterial identification must be explained. Which method was used ? For example, Pastorex kit is a very old method.

-5. The reference method to determine glycopeptides susceptibility is the microdilution (read EUCAST). The results presented in this study are not adapted.

-6. This work is a retrospective and monocentric study. The results could not be extended to specialized diabetic foot clinics or a Center in warm countries where the ecology of foot ulcers is different.

-7. Please explain why your population is so young (1/3 with less than 60 years) and composed by an important number of Type 1 diabetes (20%). This is not characteristic to the population with DFU (more than 60 years and Type 2 diabetes).

-8. Please explain the importance of Clostridium sp. in this population. Describe the different species identified in this paper.

-9. The authors must explain why MRSA would be more often isolated in Type 2 diabetes ?

-10. Discussion-P. 9: “Generally, MRSA bone and joint infections are considered more severe….” The references used concerned children infections. Some specific references concerning DFI and DFOM are not conclusive on this link between MRSA and the severity of the wounds (Hartemann-Heurtier A et al., Diabet Med 2004; Richard JL, et al., Diabetes Metab 2008; Aragon-Sanchez J et al., Diabet Med 2009). As we could have a doubt concerning the systematic use of bone biopsies in this study, this sentence must be modulated and discussed.

-11/ Finally, to definitively conclude that MRSA involved severe infections, all the patients must be followed-up and the authors must report the evolution of the wounds (e.g., number of amputations, number of death etc). Without this, no conclusion can be drawn on the severity of the presence of MRSA.

Minor points:

-There are many typographical errors along the text. Please correct (e.g., for statistical result, please used p and not q)

-Results-P. 2: Delete “and 51 (25.4%) were women”. Delete “66 (32.8%) patients were between 40 and 60 years, and 3 persons (1.5%) were under 40 years”. All these points are in the Table 1. You could modified with “69 (34.3%) patients had less than 60 years”.

-Results-P.3: Delete “such as Escherichia coli …. others”. We know the different species included in the Enterobacteriaceae family. In this paragraph (2.2. Isolated bacteria), the description of the different genus or species identified is non adapted because all the data are present in the Table 2. Please indicate the most important results with the percentage.

-Table 2: Please correct Alpha-hemolytic streptococci, Enterobacter cloacae, Acinetobacter baumannii and Veillonella parvula